# Fast and Memory Optimal Low-Rank Matrix Approximation

**Se-Young Yun**
MSR, Cambridge
seyoung.yun@inria.fr

**Marc Lelarge** [*]
Inria & ENS
marc.lelarge@ens.fr

**Alexandre Proutiere** [†]
KTH, EE School / ACL
alepro@kth.se

## Abstract

In this paper, we revisit the problem of constructing a near-optimal rank $k$ approximation of a matrix $M \in [0,1]^{m \times n}$ under the streaming data model where the columns of $M$ are revealed sequentially. We present SLA (Streaming Low-rank Approximation), an algorithm that is asymptotically accurate, when $ks_{k+1}(M) = o(\sqrt{mn})$ where $s_{k+1}(M)$ is the $(k+1)$-th largest singular value of $M$. This means that its average mean-square error converges to 0 as $m$ and $n$ grow large (i.e., $\|\hat{M}^{(k)} - M^{(k)}\|_F^2 = o(mn)$ with high probability, where $\hat{M}^{(k)}$ and $M^{(k)}$ denote the output of SLA and the optimal rank $k$ approximation of $M$, respectively). Our algorithm makes one pass on the data if the columns of $M$ are revealed in a random order, and two passes if the columns of $M$ arrive in an arbitrary order. To reduce its memory footprint and complexity, SLA uses random sparsification, and samples each entry of $M$ with a small probability $\delta$. In turn, SLA is memory optimal as its required memory space scales as $k(m+n)$, the dimension of its output. Furthermore, SLA is computationally efficient as it runs in $O(\delta kmn)$ time (a constant number of operations is made for each observed entry of $M$), which can be as small as $O(k \log(m)^4 n)$ for an appropriate choice of $\delta$ and if $n \geq m$.

## 1 Introduction

We investigate the problem of constructing, in a memory and computationally efficient manner, an accurate estimate of the optimal rank $k$ approximation $M^{(k)}$ of a large ($m \times n$) matrix $M \in [0,1]^{m \times n}$. This problem is fundamental in machine learning, and has naturally found numerous applications in computer science. The optimal rank $k$ approximation $M^{(k)}$ minimizes, over all rank $k$ matrices $Z$, the Frobenius norm $\|M - Z\|_F$ (and any norm that is invariant under rotation) and can be computed by Singular Value Decomposition (SVD) of $M$ in $O(nm^2)$ time (if we assume that $m \leq n$). For massive matrices $M$ (i.e., when $m$ and $n$ are very large), this becomes unacceptably slow. In addition, storing and manipulating $M$ in memory may become difficult. In this paper, we design a memory and computationally efficient algorithm, referred to as Streaming Low-rank Approximation (SLA), that computes a near-optimal rank $k$ approximation $\hat{M}^{(k)}$. Under mild assumptions on $M$, the SLA algorithm is *asymptotically accurate* in the sense that as $m$ and $n$ grow large, its average mean-square error converges to 0, i.e., $\|\hat{M}^{(k)} - M^{(k)}\|_F^2 = o(mn)$ with high probability (we interpret $M^{(k)}$ as the signal that we aim to recover form a noisy observation $M$).

To reduce its memory footprint and running time, the proposed algorithm combines random sparsification and the idea of the streaming data model. More precisely, each entry of $M$ is revealed to the algorithm with probability $\delta$, called the sampling rate. Moreover, SLA observes and treats the

---

[*]Work performed as part of MSR-INRIA joint research centre. M.L. acknowledges the support of the French Agence Nationale de la Recherche (ANR) under reference ANR-11-JS02-005-01 (GAP project).

[†]A. Proutiere's research is supported by the ERC FSA grant, and the SSF ICT-Psi project.

columns of $M$ one after the other in a sequential manner. The sequence of observed columns may be chosen uniformly at random in which case the algorithm requires one pass on $M$ only, or can be arbitrary in which case the algorithm needs two passes. SLA first stores $\ell = 1/(\delta \log(m))$ randomly selected columns, and extracts via spectral decomposition an estimator of parts of the $k$ top right singular vectors of $M$. It then completes the estimator of these vectors by receiving and treating the remain columns sequentially. SLA finally builds, from the estimated top $k$ right singular vectors, the linear projection onto the subspace generated by these vectors, and deduces an estimator of $M^{(k)}$. The analysis of the performance of SLA is presented in Theorems 7, and 8. In summary: when $m \leq n$, $\frac{\log^4(m)}{m} \leq \delta \leq m^{-8/9}$, with probability $1 - k\delta$, the output $\hat{M}^{(k)}$ of SLA satisfies:

$$\frac{\|M^{(k)} - \hat{M}^{(k)}\|_F^2}{mn} = O\left(k^2\left(\frac{s_{k+1}^2(M)}{mn} + \frac{\log(m)}{\sqrt{\delta m}}\right)\right), \tag{1}$$

where $s_{k+1}(M)$ is the $(k+1)$-th singular value of $M$. SLA requires $O(kn)$ memory space, and if $\delta \geq \frac{\log^4(m)}{m}$ and $k \leq \log^6(m)$, its time is $O(\delta kmn)$. To ensure the asymptotic accuracy of SLA, the upper-bound in (1) needs to converge to 0 which is true as soon as $ks_{k+1}(M) = o(\sqrt{mn})$. In the case where $M$ is seen as a noisy version of $M^{(k)}$, this condition quantifies the maximum amount of noise allowed for our algorithm to be asymptotically accurate.

SLA is memory optimal, since any rank $k$ approximation algorithm needs to at least store its output, i.e., $k$ right and left singular vectors, and hence needs at least $O(kn)$ memory space. Further observe that among the class of algorithms sampling each entry of $M$ at a given rate $\delta$, SLA is computational optimal, since it runs in $O(\delta kmn)$ time (it does a constant number of operations per observed entry if $k = O(1)$). In turn, to the best of our knowledge, SLA is both faster and more memory efficient than existing algorithms. SLA is the first memory optimal and asymptotically accurate low rank approximation algorithm.

The approach used to design SLA can be readily extended to devise memory and computationally efficient matrix completion algorithms. We present this extension in the supplementary material.

**Notations.** Throughout the paper, we use the following notations. For any $m \times n$ matrix $A$, we denote by $A^\top$ its transpose, and by $A^{-1}$ its pseudo-inverse. We denote by $s_1(A) \geq \cdots \geq s_{n \wedge m}(A) \geq 0$, the singular values of $A$. When matrices $A$ and $B$ have the same number of rows, $[A, B]$ to denote the matrix whose first columns are those of $A$ followed by those of $B$. $A_\perp$ denotes an orthonormal basis of the subspace perpendicular to the linear span of the columns of $A$. $A_j$, $A^i$, and $A_{ij}$ denote the $j$-th column of $A$, the $i$-th row of $A$, and the entry of $A$ on the $i$-th line and $j$-th column, respectively. For $h \leq l$, $A_{h:l}$ (resp. $A^{h:l}$) is the matrix obtained by extracting the columns (resp. lines) $h, \ldots, l$ of $A$. For any ordered set $B = \{b_1, \ldots, b_p\} \subset \{1, \ldots, n\}$, $A_{(B)}$ refers to the matrix composed by the ordered set $B$ of columns of $A$. $A^{(B)}$ is defined similarly (but for lines). For real numbers $a \leq b$, we define $|A|_a^b$ the matrix with $(i, j)$ entry equal to $(|A|_a^b)_{ij} = \min(b, \max(a, A_{ij}))$. Finally, for any vector $v$, $\|v\|$ denotes its Euclidean norm, whereas for any matrix $A$, $\|A\|_F$ denotes its Frobenius norm, $\|A\|_2$ its operator norm, and $\|A\|_\infty$ its $\ell_\infty$-norm, i.e., $\|A\|_\infty = \max_{i,j} |A_{ij}|$.

## 2 Related Work

Low-rank approximation algorithms have received a lot of attention over the last decade. There are two types of error estimate for these algorithms: either the error is additive or relative.

To translate our bound (1) in an additive error is easy:

$$\|M - \hat{M}^{(k)}\|_F \leq \|M - M^{(k)}\|_F + O\left(k\left(\frac{s_{k+1}(M)}{\sqrt{mn}} + \frac{\log^{1/2} m}{(\delta m)^{1/4}}\right)\sqrt{mn}\right). \tag{2}$$

Sparsifying $M$ to speed-up the computation of a low-rank approximation has been proposed in the literature and the best additive error bounds have been obtained in [AM07]. When the sampling rate $\delta$ satisfies $\delta \geq \frac{\log^4 m}{m}$, the authors show that with probability $1 - \exp(-\log^4 m)$,

$$\|M - \tilde{M}^{(k)}\|_F \leq \|M - M^{(k)}\|_F + O\left(\frac{k^{1/2}n^{1/2}}{\delta^{1/2}} + \frac{k^{1/4}n^{1/4}}{\delta^{1/4}}\|M^{(k)}\|_F^{1/2}\right). \tag{3}$$

This performance guarantee is derived from Lemma 1.1 and Theorem 1.4 in [AM07]. To compare (2) and (3), note that our assumptions on the bounded entries of $M$ ensures that: $\frac{s_{k+1}^2(M)}{mn} \leq \frac{1}{k}$ and $\|M^{(k)}\|_F \leq \|M\|_F \leq \sqrt{mn}$. In particular, we see that the worst case bound for (3) is $\left( \frac{k^{1/2}}{\sqrt{\delta m}} + \frac{k^{1/4}}{(\delta m)^{1/4}} \right) \sqrt{nm}$ which is always lower than the worst case bound for (2): $k \left( \frac{1}{k} + \frac{\log m}{\sqrt{\delta m}} \right)^{1/2} \sqrt{nm}$. When $k = O(1)$, our bound is only larger by a logarithmic term in $m$ compared to [AM07]. However, the algorithm proposed in [AM07] requires to store $O(\delta mn)$ entries of $M$ whereas SLA needs $O(n)$ memory space. Recall that $\log^4 m \leq \delta m \leq m^{1/9}$ so that our algorithm makes a significant improvement on the memory requirement at a low price in the error guarantee bounds. Although biased sampling algorithms can reduce the error, the algorithm have to run leverage scores with multiple passes over data [BJS15]. In a recent work, [CW13] proposes a time efficient algorithm to compute a low-rank approximation of a sparse matrix. Combined with [AM07], we obtain an algorithm running in time $O(\delta mn) + O(nk^2 + k^3)$ but with an increased additive error term.

We can also compare our result to papers providing an estimate $\tilde{M}^{(k)}$ of the optimal low-rank approximation of $M$ with a relative error $\varepsilon$, i.e. such that $\|M - \tilde{M}^{(k)}\|_F \leq (1+\varepsilon)\|M - M^{(k)}\|_F$. To the best of our knowledge, [CW09] provides the best result in this setting. Theorem 4.4 in [CW09] shows that provided the rank of $M$ is at least $2(k+1)$, their algorithm outputs with probability $1-\eta$ a rank-$k$ matrix $\tilde{M}^{(k)}$ with relative error $\varepsilon$ using memory space $O\left(k/\varepsilon \log(1/\eta)(n+m)\right)$ (note that in [CW09], the authors use as unit of memory a bit whereas we use as unit of memory an entry of the matrix so we removed a $\log mn$ factor in their expression to make fair comparisons). To compare with our result, we can translate our bound (1) in a relative error, and we need to take:

$$\varepsilon = O\left( k \frac{s_{k+1}(M) + \frac{\log^{1/2} m}{(\delta m)^{1/4}} \sqrt{mn}}{\|M - M^{(k)}\|_F} \right).$$

First note that since $M$ is assumed to be of rank at least $2(k+1)$, we have $\|M - M^{(k)}\|_F \geq s_{k+1}(M) > 0$ and $\varepsilon$ is well-defined. Clearly, for our $\varepsilon$ to tend to zero, we need $\|M - M^{(k)}\|_F$ to be not too small. For the scenario we have in mind, $M$ is a noisy version of the signal $M^{(k)}$ so that $M - M^{(k)}$ is the noise matrix. When every entry of $M - M^{(k)}$ is generated independently at random with a constant variance, $\|M - M^{(k)}\|_F = \Theta(\sqrt{m+n})$ while $s_{k+1}(M) = \Theta(\sqrt{n})$. In such a case, we have $\varepsilon = o(1)$ and we improve the memory requirement of [CW09] by a factor $\varepsilon^{-1} \log(k\delta)^{-1}$. [CW09] also considers a model where the full columns of $M$ are revealed one after the other in an arbitrary order, and proposes a one-pass algorithm to derive the rank-$k$ approximation of $M$ with the same memory requirement. In this general setting, our algorithm is required to make two passes on the data (and only one pass if the order of arrival of the column is random instead of arbitrary). The running time of the algorithm scales as $O(kmn\varepsilon^{-1} \log(k\delta)^{-1})$ to project $M$ onto $k\varepsilon^{-1} \log(k\delta)^{-1}$ dimensional random space. Thus, SLA improves the time again by a factor of $\varepsilon^{-1} \log(k\delta)^{-1}$.

We could also think of using sketching and streaming PCA algorithms to estimate $M^{(k)}$. When the columns arrive sequentially, these algorithms identify the left singular vectors using one-pass on the matrix and then need a second pass on the data to estimate the right singular vectors. For example, [Lib13] proposes a sketching algorithm that updates the $p$ most frequent directions as columns are observed. [GP14] shows that with $O(km/\varepsilon)$ memory space (for $p = k/\varepsilon$), this sketching algorithm finds $m \times k$ matrix $\hat{U}$ such that $\|M - P_{\hat{U}} M\|_F \leq (1+\varepsilon)\|M - M^{(k)}\|_F$, where $P_{\hat{U}}$ denotes the projection matrix to the linear span of the columns of $\hat{U}$. The running time of the algorithm is roughly $O(kmn\varepsilon^{-1})$, which is much greater than that of SLA. Note also that to identify such matrix $\hat{U}$ in one pass on $M$, it is shown in [Woo14] that we have to use $\Omega(km/\varepsilon)$ memory space. This result does not contradict the performance analysis of SLA, since the latter needs two passes on $M$ if the columns of $M$ are observed in an arbitrary manner. Finally, note that the streaming PCA algorithm proposed in [MCJ13] does not apply to our problem as this paper investigates a very specific problem: the spiked covariance model where a column is randomly generated in an i.i.d. manner.

## 3 Streaming Low-rank Approximation Algorithm

---

**Algorithm 1** Streaming Low-rank Approximation (SLA)

---

**Input:** $M$, $k$, $\delta$, and $\ell = \frac{1}{\delta \log(m)}$
1. $A_{(B_1)}, A_{(B_2)} \leftarrow$ independently sample entries of $[M_1, \ldots, M_\ell]$ at rate $\delta$
2. PCA for the first $\ell$ columns: $Q \leftarrow \text{SPCA}(A_{(B_1)}, k)$
3. Trimming the rows and columns of $A_{(B_2)}$:
    $A_{(B_2)} \leftarrow$ set the entries of rows of $A_{(B_2)}$ having more than two non-zero entries to 0
    $A_{(B_2)} \leftarrow$ set the entries of the columns of $A_{(B_2)}$ having more than $10m\delta$ non-zero entries to 0
4. $W \leftarrow A_{(B_2)}Q$    5. $\hat{V}^{(B_1)} \leftarrow (A_{(B_1)})^\top W$    6. $\hat{I} \leftarrow A_{(B_1)}\hat{V}^{(B_1)}$
Remove $A_{(B_1)}$, $A_{(B_2)}$, and $Q$ from the memory space
**for** $t = \ell + 1$ **to** $n$ **do**
    7. $A_t \leftarrow$ sample entries of $M_t$ at rate $\delta$    8. $\hat{V}^t \leftarrow (A_t)^\top W$    9. $\hat{I} \leftarrow \hat{I} + A_t \hat{V}^t$
    Remove $A_t$ from the memory space
**end for**
10. $\hat{R} \leftarrow$ find $\hat{R}$ using the Gram-Schmidt process such that $\hat{V}\hat{R}$ is an orthonormal matrix
11. $\hat{U} \leftarrow \frac{1}{\delta} \hat{I}\hat{R}\hat{R}^\top$
**Output:** $\hat{M}^{(k)} = |\hat{U}\hat{V}^\top|_0^1$

---

---

**Algorithm 2** Spectral PCA (SPCA)

---

**Input:** $C \in [0,1]^{m \times \ell}$, $k$
$\Omega \leftarrow \ell \times k$ Gaussian random matrix
Trimming: $\bar{C} \leftarrow$ set the entries of the rows of $C$ with more than 10 non-zero entries to 0
$\Phi \leftarrow \bar{C}^\top \bar{C} - \text{diag}(\bar{C}^\top \bar{C})$
Power Iteration: $QR \leftarrow$ QR decomposition of $\Phi^{\lceil 5 \log(\ell) \rceil}\Omega$
**Output:** $Q$

---

In this section, we present the Streaming Low-rank Approximation (SLA) algorithm and analyze its performance. SLA makes one pass on the matrix $M$, and is provided with the columns of $M$ one after the other in a streaming manner. The SVD of $M$ is $M = U\Sigma V^\top$ where $U$ and $V$ are $(m \times m)$ and $(n \times n)$ unitary matrices and $\Sigma$ is the $(m \times n)$ matrix $\text{diag}(s_1(M), \ldots s_{n \wedge m}(M))$. We assume (or impose by design of SLA) that the $\ell$ (specified below) first observed columns of $M$ are chosen uniformly at random among all columns. An extension of SLA to scenarios where columns are observed in an arbitrary order is presented in §3.5, but this extension requires two passes on $M$. To be memory efficient, SLA uses sampling. Each observed entry of $M$ is erased (i.e., set equal to 0) with probability $1 - \delta$, where $\delta > 0$ is referred to as the sampling rate. The algorithm, whose pseudo-code is presented in Algorithm 1, proceeds in three steps:

1. In the first step, we observe $\ell = \frac{1}{\delta \log(m)}$ columns of $M$ chosen uniformly at random. These columns form the matrix $M_{(B)} = U\Sigma(V^{(B)})^\top$, where $B$ denotes the ordered set of the indexes of the $\ell$ first observed columns. $M_{(B)}$ is sampled at rate $\delta$. More precisely, we apply two independent sampling procedures, where in each of them, every entry of $M_{(B)}$ is sampled at rate $\delta$. The two resulting independent random matrices $A_{(B_1)}$, and $A_{(B_2)}$ are stored in memory. $A_{(B_1)}$, referred to as $A_{(B)}$ to simplify the notations, is used in this first step, whereas $A_{(B_2)}$ will be used in subsequent steps. Next through a spectral decomposition of $A_{(B)}$, we derive a $(\ell \times k)$ orthonormal matrix $Q$ such that the span of its column vectors approximates that of the column vectors of $V_{1:k}^{(B)}$. The first step corresponds to Lines 1 and 2 in the pseudo-code of SLA.

2. In the second step, we complete the construction of our estimator of the top $k$ right singular vectors $V_{1:k}$ of $M$. Denote by $\hat{V}$ the $k \times n$ matrix formed by these estimated vectors. We first compute the components of these vectors corresponding to the set of indexes $B$ as $\hat{V}^{(B)} = A_{(B_1)}^\top W$ with $W = A_{(B2)}Q$. Then for $t = \ell + 1, \ldots, n$, after receiving the $t$-th column $M_t$ of $M$, we set $\hat{V}^t = A_t^\top W$, where $A_t$ is obtained by sampling entries of $M_t$ at rate $\delta$. Hence after one pass on $M$, we get $\hat{V} = \tilde{A}^\top W$, where $\tilde{A} = [A_{(B_1)}, A_{\ell+1}, \ldots, A_n]$. As it turns out, multiplying $W$ by $\tilde{A}^\top$ amplifies the useful signal contained in $W$, and yields an accurate approximation of the span of the

top $k$ right singular vectors $V_{1:k}$ of $M$. The second step is presented in Lines 3, 4, 5, 7 and 8 in SLA pseudo-code.

3. In the last step, we deduce from $\hat{V}$ a set of column vectors gathered in matrix $\hat{U}$ such that $\hat{U}^\top \hat{V}$ provides an accurate approximation of $M^{(k)}$. First, using the Gram-Schmidt process, we find $\hat{R}$ such that $\hat{V}\hat{R}$ is an orthonormal matrix and compute $\hat{U} = \frac{1}{\delta} A\hat{V}\hat{R}\hat{R}^\top$ in a streaming manner as in Step 2. Then, $\hat{U}\hat{V}^\top = \frac{1}{\delta}A\hat{V}\hat{R}(\hat{V}\hat{R})^\top$ where $\hat{V}\hat{R}(\hat{V}\hat{R})^\top$ approximates the projection matrix onto the linear span of the top $k$ right singular vectors of $M$. Thus, $\hat{U}\hat{V}^\top$ is close to $M^{(k)}$. This last step is described in Lines 6, 9, 10 and 11 in SLA pseudo-code.

In the next subsections, we present in more details the rationale behind the three steps of SLA, and provide a performance analysis of the algorithm.

## 3.1  Step 1. Estimating right-singular vectors of the first batch of columns

The objective of the first step is to estimate $V_{1:k}^{(B)}$, those components of the top $k$ right singular vectors of $M$ whose indexes are in the set $B$ (remember that $B$ is the set of indexes of the $\ell$ first observed columns). This estimator, denoted by $Q$, is obtained by applying the power method to extract the top $k$ right singular vector of $M_{(B)}$, as described in Algorithm 2. In the design of this algorithm and its performance analysis, we face two challenges: (i) we only have access to a sampled version $A_{(B)}$ of $M_{(B)}$; and (ii) $U\Sigma(V^{(B)})^\top$ is not the SVD of $M_{(B)}$ since the column vectors of $V_{1:k}^{(B)}$ are not orthonormal in general (we keep the components of these vectors corresponding to the set of indexes $B$). Hence, the top $k$ right singular vectors of $M_{(B)}$ that we extract in Algorithm 2 do not necessarily correspond to $V_{1:k}^{(B)}$.

To address (i), in Algorithm 2, we do not directly extract the top $k$ right singular vectors of $A_{(B)}$. We first remove the rows of $A_{(B)}$ with too many non-zero entries (i.e., too many observed entries from $M_{(B)}$), since these rows would perturb the SVD of $A_{(B)}$. Let us denote by $\bar{A}$ the obtained trimmed matrix. We then form the covariance matrix $\bar{A}^\top \bar{A}$, and remove its diagonal entries to obtain the matrix $\Phi = \bar{A}^\top \bar{A} - \mathrm{diag}(\bar{A}^\top \bar{A})$. Removing the diagonal entries is needed because of the sampling procedure. Indeed, the diagonal entries of $\bar{A}^\top \bar{A}$ scale as $\delta$, whereas its off-diagonal entries scale as $\delta^2$. Hence, when $\delta$ is small, the diagonal entries would clearly become dominant in the spectral decomposition. We finally apply the power method to $\Phi$ to obtain $Q$. In the analysis of the performance of Algorithm 2, the following lemma will be instrumental, and provides an upper bound of the gap between $\Phi$ and $(M_{(B)})^\top M_{(B)}$ using the matrix Bernstein inequality (Theorem 6.1 [Tro12]). All proofs are detailed in Appendix.

**Lemma 1** *If $\delta \leq m^{-\frac{8}{9}}$, with probability $1 - \frac{1}{\ell^2}$, $\|\Phi - \delta^2(M_{(B)})^\top M_{(B)}\|_2 \leq c_1 \delta\sqrt{m\ell \log(\ell)}$, for some constant $c_1 > 1$.*

To address (ii), we first establish in Lemma 2 that for an appropriate choice of $\ell$, the column vectors of $V_{1:k}^{(B)}$ are approximately orthonormal. This lemma is of independent interest, and relates the SVD of a truncated matrix, here $M_{(B)}$, to that of the initial matrix $M$. More precisely:

**Lemma 2** *If $\delta \leq m^{-8/9}$, there exists a $\ell \times k$ matrix $\bar{V}^{(B)}$ such that its column vectors are orthonormal, and with probability $1 - \exp(-m^{1/7})$, for all $i \leq k$ satisfying that $s_i^2(M) \geq \frac{n}{\delta\ell}\sqrt{m\ell \log(\ell)}$, $\|\sqrt{\frac{n}{\ell}}V_{1:i}^{(B)} - \bar{V}_{1:i}^{(B)}\|_2 \leq m^{-\frac{1}{3}}$.*

Note that as suggested by the above lemma, it might be impossible to recover $V_i^{(B)}$ when the corresponding singular value $s_i(M)$ is small (more precisely, when $s_i^2(M) \leq \frac{n}{\delta\ell}\sqrt{m\ell \log(\ell)}$). However, the singular vectors corresponding to such small singular values generate very little error for low-rank approximation. Thus, we are only interested in singular vectors whose singular values are above the threshold $(\frac{n}{\delta\ell}\sqrt{m\ell \log(\ell)})^{1/2}$. Let $k' = \max\{i : s_i^2(M) \geq \frac{n}{\delta\ell}\sqrt{m\ell \log(\ell)}, i \leq k\}$.

Now to analyze the performance of Algorithm 2 when applied to $A_{(B)}$, we decompose $\Phi$ as $\Phi = \frac{\delta^2\ell}{n}\bar{V}_{1:k'}^{(B)}(\Sigma_{1:k'}^{1:k'})^2(\bar{V}_{1:k'}^{(B)})^\top + Y$, where $Y = \Phi - \frac{\delta^2\ell}{n}\bar{V}_{1:k'}^{(B)}(\Sigma_{1:k'}^{1:k'})^2(\bar{V}_{1:k'}^{(B)})^\top$ is a *noise* matrix. The

following lemma quantifies how noise may affect the performance of the power method, i.e., it provides an upper bound of the gap between $Q$ and $\bar{V}_{1:k'}^{(B)}$ as a function of the operator norm of the noise matrix $Y$:

**Lemma 3** *With probability* $1 - \frac{1}{\ell^2}$, *the output $Q$ of SPCA when applied to $A_{(B)}$ satisfies for all* $i \leq k'$: $\|(\bar{V}_{1:i}^{(B)})^\top \cdot Q_\perp\|_2 \leq \frac{3\|Y\|_2}{\delta^2 \frac{\ell}{n} s_i(M)^2}$.

In the proof, we analyze the power iteration algorithm from results in [HMT11].

To complete the performance analysis of Algorithm 2, it remains to upper bound $\|Y\|_2$. To this aim, we decompose $Y$ into three terms:

$$Y = \left(\Phi - \delta^2 (M_{(B)})^\top M_{(B)}\right) + \delta^2 (M_{(B)})^\top \left(I - U_{1:k'} U_{1:k'}^\top\right) M_{(B)} +$$

$$\delta^2 \left((M_{(B)})^\top U_{1:k'} U_{1:k'}^\top M_{(B)} - \frac{\ell}{n} \bar{V}_{1:k'}^{(B)} (\Sigma_{1:k'}^{1:k'})^2 (\bar{V}_{1:k'}^{(B)})^\top\right).$$

The first term can be controlled using Lemma 1, and the last term is upper bounded using Lemma 2. Finally, the second term corresponds to the error made by ignoring the singular vectors which are not within the top $k'$. To estimate this term, we use the matrix Chernoff bound (Theorem 2.2 in [Tro11]), and prove that:

**Lemma 4** *With probability* $1 - \exp(-m^{1/4})$, $\|(I - U_{1:k'} U_{1:k'}^\top) M_{(B)}\|_2^2 \leq \frac{2}{\delta} \sqrt{m\ell \log(\ell)} + \frac{\ell}{n} s_{k+1}^2(M)$.

In summary, combining the four above lemmas, we can establish that $Q$ accurately estimates $\bar{V}_{1:k}^{(B)}$:

**Theorem 5** *If $\delta \leq m^{-8/9}$, with probability* $1 - \frac{3}{\ell^2}$, *the output $Q$ of Algorithm 2 when applied to $A_{(B)}$ satisfies for all* $i \leq k$: $\|(\bar{V}_{1:i}^{(B)})^\top \cdot Q_\perp\|_2 \leq \frac{3\delta^2 (s_{k+1}^2(M) + 2m^{\frac{2}{3}} n) + 3(2 + c_1)\delta\frac{n}{\ell}\sqrt{m\ell \log(\ell)}}{\delta^2 s_i^2(M)}$, *where $c_1$ is the constant from Lemma 1.*

### 3.2 Step 2: Estimating the principal right singular vectors of $M$

In this step, we aim at estimating the top $k$ right singular vectors $V_{1:k}$, or at least at producing $k$ vectors whose linear span approximates that of $V_{1:k}$. Towards this objective, we start from $Q$ derived in the previous step, and define the $(m \times k)$ matrix $W = A_{(B_2)} Q$. $W$ is stored and kept in memory for the remaining of the algorithm.

It is tempting to directly read from $W$ the top $k'$ left singular vectors $U_{1:k'}$. Indeed, we know that $Q \approx \sqrt{\frac{n}{\ell}} V_{1:k}^{(B)}$, and $\mathbb{E}[A_{(B_2)}] = \delta U \Sigma (V^{(B)})^\top$, and hence $\mathbb{E}[W] \approx \delta \sqrt{\frac{n}{\ell}} U_{1:k} \Sigma_{1:k}^{1:k}$. However, the level of the noise in $W$ is too important so as to accurately extract $U_{1:k'}$. In turn, $W$ can be written as $\delta U \Sigma (V^{(B)})^\top Q + Z$, where $Z = (A_{(B_2)} - \delta U \Sigma (V^{(B)})^\top) Q$ partly captures the noise in $W$. It is then easy to see that the level of the noise $Z$ satisfies $\mathbb{E}[\|Z\|_2] \geq \mathbb{E}[\|Z\|_F / \sqrt{k}] = \Omega(\sqrt{\delta m})$. Indeed, first observe that $Z$ is of rank $k$. Then $\mathbb{E}[\|Z\|_F^2] = \sum_{i=1}^m \sum_{j=1}^k \mathbb{E}[Z_{ij}^2] \approx mk\delta$: this is due to the facts that (i) $Q$ and $A_{(B_2)} - \delta U \Sigma (V^{(B)})^\top$ are independent (since $A_{(B_1)}$ and $A_{(B_2)}$ are independent), (ii) $\|Q_j\|_2^2 = 1$ for all $j \leq k$, and (iii) the entries of $A_{(B_2)}$ are independent with variance $\Theta(\delta(1 - \delta))$. However, for all $j \leq k'$, the $j$-th singular value of $\delta U \Sigma (V^{(B)})^\top Q$ scales as $O(\delta\sqrt{m\ell}) = O(\sqrt{\frac{\delta m}{\log(m)}})$, since $s_j(M) \leq \sqrt{mn}$ and $s_j(M_{(B)}) \approx \sqrt{\frac{\ell}{n}} s_j(M)$ when $j \leq k'$ from Lemma 2.

Instead, from $W$, $A_{(B_1)}$ and the subsequent sampled arriving columns $A_t$, $t > \ell$, we produce a $(n \times k)$ matrix $\hat{V}$ whose linear span approximates that of $V_{1:k'}$. More precisely, we first let $\hat{V}^{(B)} = A_{(B_1)}^\top W$. Then for all $t = \ell + 1, \ldots, n$, we define $\hat{V}^t = A_t^\top W$, where $A_t$ is obtained from the $t$-th observed column of $M$ after sampling each of its entries at rate $\delta$. Multiplying $W$ by $\tilde{A} = [A_{(B_1)}, A_{\ell+1}, \ldots, A_n]$ amplifies the useful signal in $W$, so that $\hat{V} = \tilde{A}^\top W$ constitutes a good approximation of $V_{1:k}$. To understand why, we can rewrite $\hat{V}$ as follows:

$$\hat{V} = \delta^2 M^\top M_{(B)} Q + \delta M^\top (A_{(B_2)} - \delta M_{(B)}) Q + (\tilde{A} - \delta M)^\top W.$$

In the above equation, the first term corresponds to the useful signal and the two remaining terms constitute noise matrices. From Theorem 5, the linear span of columns of $Q$ approximates that of the columns of $\bar{V}^{(B)}$ and thus, for $j \leq k'$, $s_j(\delta^2 M^\top M_{(B)} Q) \approx \delta^2 s_j^2(M) \sqrt{\frac{\ell}{n}} \geq \delta \sqrt{mn \log(\ell)}$. The spectral norms of the noise matrices are bounded using random matrix arguments, and the fact that $(A_{(B_2)} - \delta M_{(B)})$ and $(\tilde{A} - \delta M)$ are zero-mean random matrices with independent entries. We can show (see Lemma 14 given in the supplementary material) using the independence of $A_{(B_1)}$ and $A_{(B2)}$ that with high probability, $\|\delta M^\top (A_{(B_2)} - \delta M_{(B)})Q\|_2 = O(\delta \sqrt{mn})$. We may also establish that with high probability, $\|(\tilde{A} - \delta M)^\top W\|_2 = O(\delta \sqrt{m(m+n)})$. This is a consequence of a result derived in [AM07] (quoted in Lemma 13 in the supplementary material) stating that with high probability, $\|\tilde{A} - \delta M\| = O(\sqrt{\delta(m+n)})$ and of the fact that due to the trimming process presented in Line 3 in Algorithm 1, $\|W\|_2 = O(\sqrt{\delta m})$. In summary, as soon as $n$ scales at least as $m$, the noise level becomes negligible, and the span of $\hat{V}_{1:k'}$ provides an accurate approximation of that of $V_{1:k'}$. The above arguments are made precise and rigorous in the supplementary material. The following theorem summarizes the accuracy of our estimator of $V_{1:k}$.

**Theorem 6** With $\frac{\log^4(m)}{m} \leq \delta \leq m^{-\frac{8}{9}}$ for all $i \leq k$, there exists a constant $c_2$ such that with probability $1 - k\delta$, $\|V_i^\top (\hat{V}_{1:k})_\perp\|_2 \leq c_2 \frac{s_{k+1}^2(M) + n \log(m) \sqrt{m/\delta} + m \sqrt{n \log(m)/\delta}}{s_i^2(M)}$.

### 3.3   Step 3: Estimating the principal left singular vectors of $M$

In the last step, we estimate the principal left singular vectors of $M$ to finally derive an estimator of $M^{(k)}$, the optimal rank-$k$ approximation of $M$. The construction of this estimator is based on the observation that $M^{(k)} = U_{1:k} \Sigma_{1:k}^{1:k} V_{1:k}^\top = M P_{V_{1:k}}$, where $P_{V_{1:k}} = V_{1:k} V_{1:k}^\top$ is an $(n \times n)$ matrix representing the projection onto the linear span of the top $k$ right singular vectors $V_{1:k}$ of $M$. Hence to estimate $M^{(k)}$, we try to approximate the matrix $P_{V_{1:k}}$. To this aim, we construct a $(k \times k)$ matrix $\hat{R}$ so that the column vectors of $\hat{V}\hat{R}$ form an orthonormal basis whose span corresponds to that of the column vectors of $\hat{V}$. This construction is achieved using Gram-Schmidt process. We then approximate $P_{V_{1:k}}$ by $P_{\hat{V}} = \hat{V}\hat{R}\hat{R}^\top \hat{V}^\top$, and finally our estimator $\hat{M}^{(k)}$ of $M^{(k)}$ is $\frac{1}{\delta} \tilde{A} P_{\hat{V}}$.

The construction of $\hat{M}^{(k)}$ can be made in a memory efficient way accommodating for our streaming model where the columns of $M$ arrive one after the other, as described in the pseudo-code of SLA. First, after constructing $\hat{V}^{(B)}$ in Step 2, we build the matrix $\hat{I} = A_{(B_1)} \hat{V}^{(B)}$. Then, for $t = \ell + 1, \dots, n$, after constructing the $t$-th line $\hat{V}^t$ of $\hat{V}$, we update $\hat{I}$ by adding to it the matrix $A_t \hat{V}^t$, so that after all columns of $M$ are observed, $\hat{I} = \tilde{A}\hat{V}$. Hence we can build an estimator $\hat{U}$ of the principal left singular vectors of $M$ as $\hat{U} = \frac{1}{\delta} \hat{I} \hat{R} \hat{R}^\top$, and finally obtain $\hat{M}^{(k)} = |\hat{U}\hat{V}^\top|_0^1$.

To quantify the estimation error of $\hat{M}^{(k)}$, we decompose $M^{(k)} - \hat{M}^{(k)}$ as: $M^{(k)} - \hat{M}^{(k)} = M^{(k)}(I - P_{\hat{V}}) + (M^{(k)} - M)P_{\hat{V}} + (M - \frac{1}{\delta}\tilde{A})P_{\hat{V}}$. The first term of the r.h.s. of the above equation can be bounded using Theorem 6: for $i \leq k$, we have $s_i(M)^2 \|V_i^\top \hat{V}_\perp\| \leq z = c_2(s_{k+1}^2(M) + n \log(m) \sqrt{m/\delta} + m \sqrt{n \log(m)/\delta})$, and hence we can conclude that for all $i \leq k$, $\left\| s_i(M) U_i V_i^\top (I - P_{\hat{V}}) \right\|_F^2 \leq z$. The second term can be easily bounded observing that the matrix $(M^{(k)} - M)P_{\hat{V}}$ is of rank $k$: $\|(M^{(k)} - M)P_{\hat{V}}\|_F^2 \leq k\|(M^{(k)} - M)P_{\hat{V}}\|_2^2 \leq k\|M^{(k)} - M\|_2^2 = k s_{k+1}(M)^2$. The last term in the r.h.s. can be controlled as in the performance analysis of Step 2, and observing that $(\frac{1}{\delta}\tilde{A} - M)P_{\hat{V}}$ is of rank $k$: $\| \left( \frac{1}{\delta}\tilde{A} - M \right) P_{\hat{V}}\|_F^2 \leq k \left\| \frac{1}{\delta}\tilde{A} - M \right\|_2^2 = O(k\delta(m+n))$. It is then easy to remark that for the range of the parameter $\delta$ we are interested in, the upper bound $z$ of the first term dominates the upper bound of the two other terms. Finally, we obtain the following result (see the supplementary material for a complete proof):

**Theorem 7** When $\frac{\log^4(m)}{m} \leq \delta \leq m^{-\frac{8}{9}}$, with probability $1 - k\delta$, the output of the SLA algorithm satisfies with constant $c_3$: $\frac{\|M^{(k)} - [\hat{U}\hat{V}^\top]_0^1\|_F^2}{mn} = c_3 k^2 \left( \frac{s_{k+1}^2(M)}{mn} + \frac{\log(m)}{\sqrt{\delta}m} + \sqrt{\frac{\log(m)}{\delta n}} \right)$.

Note that if $\frac{\log^4(m)}{m} \le \delta \le m^{-\frac{8}{9}}$, then $\frac{\log(m)}{\sqrt{\delta m}} = o(1)$. Hence if $n \ge m$, the SLA algorithm provides an asymptotically accurate estimate of $M^{(k)}$ as soon as $\frac{s_{k+1}(M)^2}{mn} = o(1)$.

## 3.4 Required Memory and Running Time

**Required memory.**
*Lines 1-6 in SLA pseudo-code.* $A_{(B_1)}$ and $A_{(B_2)}$ have $O(\delta m \ell)$ non-zero entries and we need $O(\delta m \ell \log m)$ bits to store the id of these entries. Similarly, the memory required to store $\Phi$ is $O(\delta^2 m \ell^2 \log(\ell))$. Storing $Q$ further requires $O(\ell k)$ memory. Finally, $\hat{V}^{(B_1)}$ and $\hat{I}$ computed in Line 6 require $O(\ell k)$ and $O(km)$ memory space, respectively. Thus, when $\ell = \frac{1}{\delta \log m}$, this first part of the algorithm requires $O(k(m+n))$ memory.
*Lines 7-9.* Before we treat the remaining columns, $A_{(B_1)}$, $A_{(B_2)}$, and $Q$ are removed from the memory. Using this released memory, when the $t$-th column arrives, we can store it, compute $\hat{V}^t$ and $\hat{I}$, and remove the column to save memory. Therefore, we do not need additional memory to treat the remaining columns.
*Lines 10 and 11.* From $\hat{I}$ and $\hat{V}$, we compute $\hat{U}$. To this aim, the memory required is $O(k(m+n))$.

**Running time.**
*From line 1 to 6.* The SPCA algorithm requires $O(\ell k(\delta^2 m \ell + k) \log(\ell))$ floating-point operations to compute $Q$. $W$, $\hat{V}$, and $\hat{I}$ are inner products, and their computations require $O(\delta k m \ell)$ operations. With $\ell = \frac{1}{\delta \log(m)}$, the number of operations to treat the first $\ell$ columns is $O(\ell k(\delta^2 m \ell + k) \log(\ell) + k \delta m \ell) = O(km) + O(\frac{k^2}{\delta})$.
*From line 7 to 9.* To compute $\hat{V}^t$ and $\hat{I}$ when the $t$-th column arrives, we need $O(\delta km)$ operations. Since there are $n - \ell$ remaining columns, the total number of operations is $O(\delta kmn)$.
*Lines 10 and 11* $\hat{R}$ is computed from $\hat{V}$ using the Gram-Schmidt process which requires $O(k^2 m)$ operations. We then compute $\hat{I}\hat{R}\hat{R}^\top$ using $O(k^2 m)$ operations. Hence we conclude that:

In summary, we have shown that:

**Theorem 8** *The memory required to run the SLA algorithm is $O(k(m+n))$. Its running time is $O(\delta kmn + \frac{k^2}{\delta} + k^2 m)$.*

Observe that when $\delta \ge \max(\frac{(\log(m))^4}{m}, \frac{(\log(m))^2}{n})$ and $k \le (\log(m))^6$, we have $\delta kmn \ge k^2/\delta \ge k^2 m$, and therefore, the running time of SLA is $O(\delta kmn)$.

## 3.5 General Streaming Model

SLA is a one-pass low-rank approximation algorithm, but the set of the $\ell$ first observed columns of $M$ needs to be chosen uniformly at random. We can readily extend SLA to deal with scenarios where the columns of $M$ can be observed in an arbitrary order. This extension requires two passes on $M$, but otherwise performs exactly the same operations as SLA. In the first pass, we extract a set of $\ell$ columns chosen uniformly at random, and in the second pass, we deal with all other columns. To extract $\ell$ randomly selected columns in the first pass, we proceed as follows. Assume that when the $t$-th column of $M$ arrives, we have already extracted $l$ columns. Then the $t$-th column is extracted with probability $\frac{\ell - l}{n - t + 1}$. This two-pass version of SLA enjoys the same performance guarantees as those of SLA.

## 4 Conclusion

This paper revisited the low rank approximation problem. We proposed a streaming algorithm that samples the data and produces a near optimal solution with a vanishing mean square error. The algorithm uses a memory space scaling linearly with the ambient dimension of the matrix, i.e. the memory required to store the output alone. Its running time scales as the number of sampled entries of the input matrix. The algorithm is relatively simple, and in particular, does exploit elaborated techniques (such as sparse embedding techniques) recently developed to reduce the memory requirement and complexity of algorithms addressing various problems in linear algebra.

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
