[Supplementary Material]

# Supplementary Material:
# Fast and Memory Optimal Low-Rank Matrix Approximation

## A Proofs

### A.1 Proof of Lemma 1

We first recall the Matrix Bernstein inequality (Theorem 6.1 [Tro12]), a concentration inequality for the sum of zero mean random matrices. We will apply this inequality to $\Phi$.

**Proposition 9 (Matrix Bernstein)** *Consider a finite set* $\{X^{(i)}\}_{1 \leq i \leq m}$ *of independent random matrices, where every* $X^{(i)}$ *is self-adjoint with dimension* $n$, $\mathbb{E}[X^{(i)}] = 0$, *and* $\|X^{(i)}\|_2 \leq R$ *almost surely. Let* $\rho^2 = \|\sum_{i=1}^{m} \mathbb{E}[X^{(i)}X^{(i)}]\|_2$. *Then,*

$$\mathbb{P}\{\|\sum_{i=1}^{m} X^{(i)}\|_2 \geq x\} \leq n \exp\left(\frac{-x^2/2}{\rho^2 + Rx/3}\right).$$

With a slight abuse of notation, in the remaining of this proof, we use $A$ instead of $A_{(B)}$. Recall that $A^i$ is the $i$-th low of $A$ and

$$\Phi - \mathbb{E}[\Phi] = \sum_{i=1}^{m} \left((A^i)^\top A^i - \mathrm{diag}((A^i)^\top A^i) - \mathbb{E}[(A^i)^\top A^i - \mathrm{diag}((A^i)^\top A^i)]\right).$$

Let $X^{(i)} = (A^i)^\top A^i - \mathrm{diag}((A^i)^\top A^i) - \mathbb{E}[(A^i)^\top A^i - \mathrm{diag}((A^i)^\top A^i)]$. Then $X^{(i)}$ is a self-adjoint $\ell \times \ell$ matrix and $\mathbb{E}[X^{(i)}] = 0$.

In order to use the Matrix Bernstein inequality, we find upper bounds of $\|X^{(i)}\|_2$ and $\rho^2 = \|\sum_{i=1}^{m} \mathbb{E}[X^{(i)}X^{(i)}]\|_2$. Since every entry of $A^i$ is independently sampled with probability $\delta$, $[X^{(i)}]_{uv} = O(1)$ if both $u$ and $v$ are sampled in $A^i$ and $O(\delta^2)$ otherwise. Therefore, since the number of non-zero entries of $A^i$ is bounded by 10, every row $u$ of $X^{(i)}$ satisfies, for all $1 \leq i \leq m$:

$$r_u = \sum_{v \neq u} |[X^{(i)}]_{uv}| = O(1) + O(\ell\delta^2) = O(1).$$

From the Gershgorin circle theorem, for all $1 \leq i \leq m$

$$\|X^{(i)}\|_2 = O(1). \tag{4}$$

To derive a bound for $\rho^2$, we need to bound the absolute value of each element of $\mathbb{E}[X^{(i)}X^{(i)}]$. Since the number of non-zero entries of $A^i$ is bounded by 10, we have

$$|\mathbb{E}[X^{(i)}X^{(i)}]_{uv}| = O(\delta^2) \quad \text{for all} \quad u \neq v \quad \text{and}$$
$$|\mathbb{E}[X^{(i)}X^{(i)}]_{uu}| = O(\delta^2\ell) \quad \text{for all} \quad 1 \leq u \leq \ell.$$

Again, from the Gershgorin circle theorem, we deduce that:

$$\rho^2 = \left\| \sum_{i=1}^m \mathbb{E}[X^{(i)} X^{(i)}] \right\|_2 = O(\delta^2 m \ell). \tag{5}$$

Therefore, from (4), (5), and Proposition 9, with probability $1 - \frac{1}{\ell^2}$,

$$\|\Phi - \mathbb{E}[\Phi]\|_2 = O\left( \sqrt{\delta^2 m \ell \log(\ell)} \right),$$

from which we conclude the proof as follows:

$$
\begin{aligned}
\|\Phi - \delta^2 (M_{(B)})^\top M_{(B)}\|_2 &\leq \|\delta^2 \mathrm{diag}((M_{(B)})^\top M_{(B)})\|_2 + \|\Phi - \mathbb{E}[\Phi]\|_2 \\
&= O(\delta \sqrt{m \ell \log(\ell)}).
\end{aligned}
$$

## A.2   Proof of Lemma 2

The proof exploits Lemma 3.4 [Tro11], quoted below.

**Lemma 10 (Corollary of Lemma 3.4 in [Tro11])** *Let $V$ be an $n \times i$ matrix with orthonormal columns and define $\rho = n \max_{1 \leq j \leq n} \|V^j\|_2^2$ and $\alpha = \frac{\ell}{\rho \log i}$. When the rows of $V$ are randomly shuffled, with probability*

$$1 - i \left( \frac{e^{-\varepsilon}}{(1-\varepsilon)^{1-\varepsilon}} \right)^{\alpha \log i} - i \left( \frac{e^{\varepsilon}}{(1+\varepsilon)^{1+\varepsilon}} \right)^{\alpha \log i}$$

*there exists an $\ell \times i$ matrix with orthonormal columns $\bar{V}$ such that*

$$\left\| V^{1:\ell} - \sqrt{\frac{\ell}{n}} \bar{V} \right\|_2 \leq \varepsilon \sqrt{\frac{\ell}{n}}.$$

In Lemma 10, we can write $\frac{e^{-\varepsilon}}{(1-\varepsilon)^{1-\varepsilon}}$ and $\frac{e^{\varepsilon}}{(1+\varepsilon)^{1+\varepsilon}}$ as functions of $\varepsilon^2$. Since $\frac{d}{dx}(-x - (1-x)\log(1-x)) = \log(1-x)$ and $\log(1-x)$ is a decreasing function,

$$
\begin{aligned}
\frac{e^{-\varepsilon}}{(1-\varepsilon)^{1-\varepsilon}} &= \exp\left(-\varepsilon - (1-\varepsilon)\log(1-\varepsilon)\right) \\
&\leq \exp\left( \frac{\varepsilon}{2} \log(1 - \frac{\varepsilon}{2}) \right) \\
&\leq \exp\left( -\frac{\varepsilon^2}{4} \right). \tag{6}
\end{aligned}
$$

Analogously, since $\frac{d}{dx}(x - (1+x)\log(1+x)) = -\log(1+x)$ and $-\log(1+x)$ is a decreasing function,

$$
\begin{aligned}
\frac{e^{\varepsilon}}{(1+\varepsilon)^{1+\varepsilon}} &= \exp\left(\varepsilon - (1+\varepsilon)\log(1+\varepsilon)\right) \\
&\leq \exp\left( -\frac{\varepsilon}{2} \log(1 + \frac{\varepsilon}{2}) \right) \\
&\leq \exp\left( -\frac{\varepsilon^2}{4(1+\varepsilon)} \right). \tag{7}
\end{aligned}
$$

Next, we evaluate the parameter $\alpha$ defined in Lemma 10 for our matrix $V_{1:i}$ and then use (6), (7), and Lemma 10 applied to $V_{1:i}$ with $\varepsilon = m^{-\frac{1}{3}}$ to prove Lemma 2. Since $M$ is a bounded matrix, every column of $M$ satisfies:

$$\sum_{j=1}^k s_j^2(M) V_{vj}^2 \leq \|M_v\|_2^2 \leq m.$$

Therefore, when $s_i^2(M) \geq \frac{n}{\delta\ell}\sqrt{m\ell\log(\ell)}$, $\rho \leq \frac{mn}{s_i^2(M)} \leq \frac{\delta m\ell}{\sqrt{m\ell\log(\ell)}}$ and thus,

$$\alpha \geq \frac{\sqrt{m\ell\log(\ell)}}{m\delta\log i} \geq \frac{1}{m^{\frac{1}{2}}\delta^{\frac{3}{2}}\log(m)}. \tag{8}$$

Let $\varepsilon = m^{-\frac{1}{3}}$. Combining (6), (7), and (8) and the result of Lemma 10, we can conclude that when $\delta \leq m^{-8/9}$, with probability $1 - \exp(-m^{1/7})$, there exists an $\ell \times i$ matrix with orthonormal columns $\bar{V}$ such that

$$\left\| V_{1:i}^{(B)} - \sqrt{\frac{\ell}{n}}\bar{V}_{1:i} \right\|_2 \leq m^{-\frac{1}{3}}\sqrt{\frac{\ell}{n}}.$$

### A.3 Proof of Lemma 3

The SPCA (Spectral Principal Component Analysis) algorithm is inspired by the randomized power iteration algorithm (Algorithm 4.3 in [HMT11]). Lemma 11 is an extension of results in [HMT11] where we show that $5\log(\ell)$ iterations are sufficient to compute the low-rank approximation of $\Phi$.

**Lemma 11** *After the power method with $5\log(\ell)$ iterations, with probability $1 - \frac{1}{\ell^2}$,*

$$\|(I - QQ^\top)\Phi\|_2 \leq 2s_{k+1}(\Phi).$$

**Proof:** Let $\Phi = U^{(\Phi)}\Sigma^{(\Phi)}(U^{(\Phi)})^\top$ be the singular value decomposition of $\Phi$ and $U_{1:k}^{(\Phi)}$ be the top $k$ singular vectors of $\Phi$. From Edelman's theorem [Ede88], the $\ell \times k$ Gaussian random matrix $\Omega$ whose entries are independent gaussian random values with unit variance satisfies:

$$\mathbb{P}\{s_k((U_{1:k}^{(\Phi)})^\top\Omega) \leq \varepsilon k^{-\frac{1}{2}}\} = O(\varepsilon). \tag{9}$$

From Proposition 2.4 in [RV10], we can bound the largest singular value of $\Omega$ as follows:

$$\mathbb{P}\{s_1(\Omega) \geq \log(\ell)\sqrt{\ell}\} \leq \exp(-\ell). \tag{10}$$

Then, from (9) and (10), with probability $1 - \frac{1}{\ell^2}$,

$$\frac{s_1(\Omega)}{s_k((U_{1:k}^{(\Phi)})^\top\Omega)} \leq \ell^4. \tag{11}$$

When the inital matrix $\Omega$ satisfies (11), from Theorem 9.1 and Theorem 9.2 in [HMT11],

$$\begin{aligned}\|(I - QQ^\top)\Phi\|_2 &\leq \left(1 + \left(\frac{s_1(\Omega)}{s_k((U_{1:k}^{(\Phi)})^\top\Omega)}\right)^2\right)^{\frac{1}{10\log\ell}} s_{k+1}(\Phi) \\ &\leq 2s_{k+1}(\Phi).\end{aligned}$$

∎

Since $\Phi = \frac{\delta^2\ell}{n}\bar{V}_{1:k'}^{(B)}(\Sigma_{1:k'}^{1:k'})^2(\bar{V}_{1:k'}^{(B)})^\top + Y$ and $s_{k+1}(\bar{V}_{1:k'}^{(B)}(\Sigma_{1:k'}^{1:k'})^2(\bar{V}_{1:k'}^{(B)})^\top) = 0$, from Lemma 11, the output $Q$ of the SPCA satisfies with probability $1 - \frac{1}{\ell^2}$

$$\begin{aligned}\|(I - QQ^\top)\Phi\|_2 &\leq 2s_{k+1}(\Phi) \\ &\leq 2s_1(Y).\end{aligned} \tag{12}$$

Therefore,

$$\begin{aligned}\|\frac{\delta^2\ell}{n}\bar{V}_{1:k'}^{(B)}(\Sigma_{1:k'}^{1:k'})^2(\bar{V}_{1:k'}^{(B)})^\top - QQ^\top\Phi\|_2 &\leq \|\frac{\delta^2\ell}{n}\bar{V}_{1:k'}^{(B)}(\Sigma_{1:k'}^{1:k'})^2(\bar{V}_{1:k'}^{(B)})^\top - \Phi\|_2 + \|\Phi - QQ^\top\Phi\|_2 \\ &\leq \|Y\|_2 + 2s_{k+1}(\Phi) \leq 3\|Y\|_2,\end{aligned}$$

from which we conclude

$$\|(\bar{V}_{1:i}^{(B)})^\top Q_\perp\|_2 \leq \frac{\|\frac{\delta^2\ell}{n}\bar{V}_{1:k'}^{(B)}(\Sigma_{1:k'}^{1:k'})^2(\bar{V}_{1:k'}^{(B)})^\top - QQ^\top\Phi\|_2}{\delta^2\frac{\ell}{n}s_i(M)^2}$$

$$\leq \frac{3\|Y\|_2}{\delta^2 \frac{\ell}{n} s_i(M)^2},$$

since

$$\|Q_\perp^\top \bar{V}_{1:i}^{(B)}\|_2 \frac{\delta^2 \ell}{n} s_i(M)^2 \leq \|\frac{\delta^2 \ell}{n} Q_\perp^\top \bar{V}_{1:i}^{(B)} (\Sigma_{1:i}^{1:i})^2 (\bar{V}_{1:i}^{(B)})^\top\|_2$$

$$\overset{(a)}{=} \|Q_\perp^\top \left( QQ^\top \Phi - \frac{\delta^2 \ell}{n} \bar{V}_{1:k'}^{(B)} (\Sigma_{1:k'}^{1:k'})^2 (\bar{V}_{1:k'}^{(B)})^\top \right) \bar{V}_{1:i}^{(B)}\|_2$$

$$\leq \|QQ^\top \Phi - \frac{\delta^2 \ell}{n} \bar{V}_{1:k'}^{(B)} (\Sigma_{1:k'}^{1:k'})^2 (\bar{V}_{1:k'}^{(B)})^\top\|_2,$$

where $(a)$ stems from the following equations:

$$0 = Q_\perp^\top QQ^\top \Phi$$

$$= Q_\perp^\top \left( \frac{\delta^2 \ell}{n} \bar{V}_{1:k'}^{(B)} (\Sigma_{1:k'}^{1:k'})^2 (\bar{V}_{1:k'}^{(B)})^\top + \left( QQ^\top \Phi - \frac{\delta^2 \ell}{n} \bar{V}_{1:k'}^{(B)} (\Sigma_{1:k'}^{1:k'})^2 (\bar{V}_{1:k'}^{(B)})^\top \right) \right)$$

$$= Q_\perp^\top \left( \frac{\delta^2 \ell}{n} \bar{V}_{1:k'}^{(B)} (\Sigma_{1:k'}^{1:k'})^2 (\bar{V}_{1:k'}^{(B)})^\top + \left( QQ^\top \Phi - \frac{\delta^2 \ell}{n} \bar{V}_{1:k'}^{(B)} (\Sigma_{1:k'}^{1:k'})^2 (\bar{V}_{1:k'}^{(B)})^\top \right) \right) \bar{V}_{1:i}^{(B)}$$

$$= \frac{\delta^2 \ell}{n} Q_\perp^\top \bar{V}_{1:i}^{(B)} (\Sigma_{1:i}^{1:i})^2 (\bar{V}_{1:i}^{(B)})^\top + Q_\perp^\top \left( QQ^\top \Phi - \frac{\delta^2 \ell}{n} \bar{V}_{1:k'}^{(B)} (\Sigma_{1:k'}^{1:k'})^2 (\bar{V}_{1:k'}^{(B)})^\top \right) \bar{V}_{1:i}^{(B)}.$$

## A.4  Proof of Lemma 4

We first recall the matrix Chernoff bound (Theorem 2.2 in [Tro11]) which provides an upper bound on the largest singular value of a sum of matrices which are randomly sampled from a matrix set without replacement.

**Proposition 12 (Matrix Chernoff)** *Let $\mathcal{X}$ be a finite set of positive-semidefinite matrices with dimension $d$ and satisfy $\max_{X \in \mathcal{X}} s_1(X) \leq \alpha$. Let $\beta_{\max} = \frac{\ell}{|\mathcal{X}|} s_1(\sum_{X \in \mathcal{X}} X)$. When $\{X^{(1)}, \ldots, X^{(\ell)}\}$ are sampled uniformly at random from $\mathcal{X}$ without replacement,*

$$\mathbb{P}\left\{ s_1(\sum_{i=1}^{\ell} X^{(i)}) \geq (1+\varepsilon)\beta_{\max} \right\} \leq d \left( \frac{e^\varepsilon}{(1+\varepsilon)^{1+\varepsilon}} \right)^{\beta_{\max}/\alpha} \quad \textit{for } \forall \varepsilon \geq 0.$$

Next we prove Lemma 4. Let $G = (I - U_{1:k'} U_{1:k'}^\top) M_{(B)}$. Indeed, $GG^\top$ is a sum of matrices sampled uniformly at random without replacement from

$$\mathcal{X} = \{ (I - U_{1:k'} U_{1:k'}^\top) M_1 ((I - U_{1:k'} U_{1:k'}^\top) M_1)^\top, \ldots, (I - U_{1:k'} U_{1:k'}^\top) M_n ((I - U_{1:k'} U_{1:k'}^\top) M_n)^\top \}.$$

We apply Chernoff bound to $GG^\top$: the dimension is $m$, $\alpha = m$ and $\beta_{\max} = \frac{\ell}{n} s_{k'+1}^2(M)$. Then, from Proposition 12,

$$\mathbb{P}\left\{ s_1(GG^\top) \geq (1+\varepsilon) \frac{\ell}{n} s_{k'+1}^2(M) \right\} \leq \ell \left( \frac{e^\varepsilon}{(1+\varepsilon)^{1+\varepsilon}} \right)^{\frac{\ell}{mn} s_{k'+1}^2(M)} \quad \text{for } \varepsilon \geq 0. \qquad (13)$$

When we set $\varepsilon^\star = \frac{n\sqrt{m\ell \log(\ell)}}{\delta \ell s_{k'+1}^2(M)}$ and $\delta \leq m^{-\frac{8}{9}}$, from (7) and (13), we get:

$$\mathbb{P}\left\{ s_1(GG^\top) \geq \frac{\ell}{n} s_{k'+1}^2(M) + \frac{1}{\delta}\sqrt{m\ell \log(\ell)} \right\} \leq \ell \exp\left( (\varepsilon^\star - (1+\varepsilon^\star)\log(1+\varepsilon^\star)) \frac{\ell}{mn} s_{k'+1}^2(M) \right)$$

$$\leq \ell \exp\left( \left( -\frac{(\varepsilon^\star)^2}{4(1+\varepsilon^\star)} \right) \frac{\ell}{mn} s_{k'+1}^2(M) \right)$$

$$\leq \exp\left( -m^{1/4} \right),$$

where the last inequality stems from the fact that when $\varepsilon^\star \geq 1$,

$$\left(\frac{(\varepsilon^\star)^2}{4(1+\varepsilon^\star)}\right)\frac{\ell}{mn}s^2_{k'+1}(M) \geq \left(-\frac{\varepsilon^\star}{8}\right)\frac{\ell}{mn}s^2_{k'+1}(M) = \frac{\sqrt{m\ell\log(\ell)}}{8\delta m} > m^{1/4} \text{ and}$$

when $\varepsilon^\star \leq 1$,

$$\left(\frac{(\varepsilon^\star)^2}{4(1+\varepsilon^\star)}\right)\frac{\ell}{mn}s^2_{k'+1}(M) \geq \left(-\frac{(\varepsilon^\star)^2}{8}\right)\frac{\ell}{mn}s^2_{k'+1}(M) = \frac{n\log(\ell)}{8\delta^2 s^2_{k'+1}(M)} > m^{1/4}.$$

From the definition of $k'$, we deduce that with probability $1 - \exp(-m^{1/4})$,

$$\|(I - U_{1:k'}U_{1:k'}^\top)M_{(B)}\|_2^2 \leq \frac{2}{\delta}\sqrt{m\ell\log(\ell)} + \frac{\ell}{n}s^2_{k+1}(M).$$

### A.5   Proof of Theorem 5

Recall that $k' = \max\{i : s_i^2(M) \geq \frac{n}{\delta\ell}\sqrt{m\ell\log(\ell)}, i \leq k\}$. We first consider $j > k'$. Since $\delta^2 s_j^2(M) < \frac{\delta n}{\ell}\sqrt{m\ell\log(\ell)}$ for all $j > k'$, it suffices to show that $\|(\bar{V}_{1:j}^{(B)})^\top Q_\perp\|_2 \leq 3$ which is trivial since $\|\bar{V}_{1:j}^{(B)}\|_2 = 1$, $\|Q_\perp\|_2 = 1$.

Consider now $i \leq k'$. From Lemma 3, with probability $1 - \frac{1}{\ell^2}$,

$$\|(\bar{V}_{1:i}^{(B)})^\top Q_\perp\|_2 \leq \frac{3\|Y\|_2}{\frac{\delta^2\ell}{n}s^2(M)} \quad \text{where} \quad \|Y\|_2 = \|\Phi - \frac{\delta^2\ell}{n}\bar{V}_{1:k'}^{(B)}(\Sigma_{1:k'}^{1:k'})^2(\bar{V}_{1:k'}^{(B)})^\top\|_2.$$

Therefore, we can conclude the proof if we show that:

$$\|\Phi - \frac{\delta^2\ell}{n}\bar{V}_{1:k'}^{(B)}(\Sigma_{1:k'}^{1:k'})^2(\bar{V}_{1:k'}^{(B)})^\top\|_2 \leq \frac{\delta^2\ell}{n}s^2_{k+1}(M) + \delta^2 m^{\frac{2}{3}}\ell + \delta\sqrt{c_1 m\ell\log(\ell)}. \qquad (14)$$

To this aim, we first split $\Phi - \frac{\delta^2\ell}{n}\bar{V}_{1:k'}^{(B)}(\Sigma_{1:k'}^{1:k'})^2(\bar{V}_{1:k'}^{(B)})^\top$ into 3 parts as follows:

$$\Phi - \frac{\delta^2\ell}{n}\bar{V}_{1:k'}^{(B)}(\Sigma_{1:k'}^{1:k'})^2(\bar{V}_{1:k'}^{(B)})^\top = \left(\Phi - \delta^2(M_{(B)})^\top M_{(B)}\right) + \delta^2(M_{(B)})^\top\left(I - U_{1:k'}U_{1:k'}^\top\right)M_{(B)} +$$
$$\delta^2\left((M_{(B)})^\top U_{1:k'}U_{1:k'}^\top M_{(B)} - \frac{\ell}{n}\bar{V}_{1:k'}^{(B)}(\Sigma_{1:k'}^{1:k'})^2(\bar{V}_{1:k'}^{(B)})^\top\right). \qquad (15)$$

Then, from Lemma 1, the first term in the r.h.s.of the above expression satisfies that with probability $1 - \frac{1}{\ell^2}$,

$$\|\Phi - \delta^2(M_{(B)})^\top M_{(B)}\|_2 \leq c_1\delta\sqrt{m\ell\log(\ell)}. \qquad (16)$$

For the second term in the r.h.s., since $\left(I - U_{1:k'}U_{1:k'}^\top\right)$ is a projection matrix,

$$\left(I - U_{1:k'}U_{1:k'}^\top\right) = \left(I - U_{1:k'}U_{1:k'}^\top\right)^\top\left(I - U_{1:k'}U_{1:k'}^\top\right),$$

we deduce, from the definition of $k'$ and Lemma 4, that with probability $1 - \exp(-m^{1/4})$,

$$\delta^2\|(M_{(B)})^\top\left(I - U_{1:k'}U_{1:k'}^\top\right)M_{(B)}\|_2 = \delta^2\|\left(I - U_{1:k'}U_{1:k'}^\top\right)M_{(B)}\|_2^2$$
$$\leq \delta^2\frac{\ell}{n}s^2_{k+1}(M) + 2\delta\sqrt{m\ell\log(\ell)}. \qquad (17)$$

Finally, from Lemma 2, with probability $1 - \exp(-m^{1/7})$,

$$\delta^2\left\|(M_{(B)})^\top U_{1:k'}U_{1:k'}^\top M_{(B)} - \frac{\ell}{n}\bar{V}_{1:k'}^{(B)}(\Sigma_{1:k'}^{1:k'})^2(\bar{V}_{1:k'}^{(B)})^\top\right\|_2$$

$$= \delta^2\left\|V_{1:k'}^{(B)}(\Sigma_{1:k'}^{1:k'})^2(V_{1:k'}^{(B)})^\top - \frac{\ell}{n}\bar{V}_{1:k'}^{(B)}(\Sigma_{1:k'}^{1:k'})^2(\bar{V}_{1:k'}^{(B)})^\top\right\|_2$$

$$\leq \delta^2\left\|V_{1:k'}^{(B)}(\Sigma_{1:k'}^{1:k'})^2(V_{1:k'}^{(B)})^\top - \sqrt{\frac{\ell}{n}}V_{1:k'}^{(B)}(\Sigma_{1:k'}^{1:k'})^2(\bar{V}_{1:k'}^{(B)})^\top\right\|_2 +$$

$$\delta^2 \left\| \sqrt{\frac{\ell}{n}} V_{1:k'}^{(B)} (\Sigma_{1:k'}^{1:k'})^2 (\bar{V}_{1:k'}^{(B)})^\top - \frac{\ell}{n} \bar{V}_{1:k'}^{(B)} (\Sigma_{1:k'}^{1:k'})^2 (\bar{V}_{1:k'}^{(B)})^\top \right\|_2$$

$$\leq \delta^2 \left\| V_{1:k'}^{(B)} (\Sigma_{1:k'}^{1:k'})^2 \right\|_2 \left\| (V_{1:k'}^{(B)})^\top - \sqrt{\frac{\ell}{n}} (\bar{V}_{1:k'}^{(B)})^\top \right\|_2 +$$

$$\delta^2 \left\| \sqrt{\frac{n}{\ell}} V_{1:k'}^{(B)} - \bar{V}_{1:k'}^{(B)} \right\|_2 \left\| \frac{\ell}{n} (\Sigma_{1:k'}^{1:k'})^2 (\bar{V}_{1:k'}^{(B)})^\top \right\|_2$$

$$\leq 2\delta^2 m^{\frac{2}{3}} \ell, \tag{18}$$

since $\|M\|_2 \leq \sqrt{mn}$, $\|V_{1:k'}^{(B)} \Sigma_{1:k'}^{1:k'}\|_2 \leq \sqrt{m\ell}$, and $\left\| \frac{\ell}{n} (\Sigma_{1:k'}^{1:k'})^2 (\bar{V}_{1:k'}^{(B)})^\top \right\|_2 \leq m^{-\frac{1}{3}}$.

Therefore, (14) holds with probability $1 - \frac{3}{\ell^2}$, when we combine (16), (17), and (18) with (15).

## A.6   Proof of Theorem 6

We first state some useful lemmas on the random matrices $\tilde{A} - \delta M$ and $A_{(B_2)} - \delta M_{(B)}$.

**Lemma 13 (Theorem 3.1 in [AM07])** *When $\frac{\log^4(m)}{m} \leq \delta$, with probability $1 - \exp(-\log^4 m)$,*

$$\|\tilde{A} - \delta M\|_2 = O(\sqrt{\delta(m+n)}).$$

**Lemma 14** *With probability $1 - \frac{k\delta}{2}$, $\|\delta M^\top (A_{(B_2)} - \mathbb{E}[A_{(B_2)}]) Q_{1:k}\|_2 \leq \sqrt{2\delta^2 mn}$.*

**Proof:** Since entries of $A_{(B_2)}$ are randomly sampled with probability $\delta$ and independent with $Q$, for all $1 \leq i \leq n$ and $1 \leq j \leq k$,

$$\mathbb{E}\left[ \left( [\delta M^\top (A_{(B_2)} - \mathbb{E}[A_{(B_2)}]) Q]_{ij} \right)^2 \right]$$

$$= \mathbb{E}\left[ \left( \delta \sum_{u=1}^m \sum_{v=1}^\ell M_{ui} [A_{(B_2)} - \mathbb{E}[A_{(B_2)}]]_{uv} Q_{vj} \right)^2 \right]$$

$$= \delta^2 \sum_{u=1}^m \sum_{v=1}^\ell M_{ui}^2 Q_{vj}^2 \mathbb{E}[([A_{(B_2)} - \mathbb{E}[A_{(B_2)}]]_{uv})^2]$$

$$\leq \delta^2 \sum_{u=1}^m M_{ui}^2 \sum_{v=1}^\ell Q_{vj}^2 \delta \leq \delta^3 m.$$

From the above inequality, $\mathbb{E}[\|\delta M^\top (A_{(B_2)} - \mathbb{E}[A_{(B_2)}]) Q\|_F^2] \leq \delta^3 kmn$. Therefore, by the Markov inequality,

$$\mathbb{P}\left\{ \|\delta M^\top (A_{(B_2)} - \mathbb{E}[A_{(B_2)}]) Q_{1:k}\|_2^2 \geq 2\delta^2 mn \right\} \leq \frac{\mathbb{E}[\|\delta M^\top (A_{(B_2)} - \mathbb{E}[A_{(B_2)}]) Q\|_F^2]}{2\delta^2 mn}$$

$$\leq k\delta.$$

∎

To prove Theorem 6, we use (19) to (23), which hold with probability $1 - k\delta$ from Lemma 2, Lemma 4, Theorem 5, Lemma 13, and Lemma 14.

$$\|\sqrt{\frac{n}{\ell}} V_{1:i}^{(B)} - \bar{V}_{1:i}^{(B)}\|_2 \leq m^{-\frac{1}{3}}, \tag{19}$$

$$\|(I - U_{1:k'} U_{1:k'}^\top) M_{(B)}\|_2^2 \leq \frac{2}{\delta} \sqrt{m\ell \log(\ell)} + \frac{\ell}{n} s_{k+1}^2(M), \tag{20}$$

$$\|(\bar{V}_{1:i}^{(B)})^\top \cdot Q_\perp\|_2 \leq \frac{3\delta^2 (s_{k+1}^2(M) + 2m^{\frac{2}{3}} n) + 3(2 + c_1)\delta \frac{n}{\ell} \sqrt{m\ell \log(\ell)}}{\delta^2 s_i^2(M)}, \tag{21}$$

$$\|\tilde{A} - \delta M\|_2 = O(\sqrt{\delta(m+n)}), \quad \text{and} \tag{22}$$

$$\|(\mathbb{E}[\tilde{A}])^\top((A_{(B_2)} - \mathbb{E}[A_{(B_2)}])Q_{1:k})\|_2 \le \sqrt{2\delta^2 mn}. \tag{23}$$

Now since $\|V_i^\top \hat{V}_\perp\|_2 = \sqrt{1 - \|V_i^\top P_{\hat{V}}\|_2} = s_k((I - V_i V_i^\top)P_{\hat{V}})$,

$$\|V_i^\top \hat{V}_\perp\|_2 = \inf_{x:\|x\|_2 > 0} \frac{\|(I - V_i V_i^\top)\hat{V}x\|_2}{\|\hat{V}x\|_2},$$

to complete the proof of Theorem 6, we need to show that for all $i \le k'$, there exists a unit vector $x^{(i)}$ such that:

$$\frac{\|(I - V_i V_i^\top)\hat{V}x^{(i)}\|_2}{\|\hat{V}x^{(i)}\|_2} = O\left(\frac{s_{k+1}^2(M) + n\log(m)\sqrt{m/\delta} + m\sqrt{n\log(m)/\delta}}{s_i^2(M)}\right). \tag{24}$$

To this aim, for all $i \le k'$, we set $x^{(i)}$ as a $\ell \times 1$ unit vector (i.e., $\|x^{(i)}\|_2 = 1$) such that

$$\|(\bar{V}_i^{(B)})^\top Q x^{(i)}\|_2 \ge \sqrt{1 - \|(\bar{V}_{1:i}^{(B)})^\top Q_\perp x^{(i)}\|_2^2}, \tag{25}$$

$$\|(\bar{V}_{i+1:k}^{(B)})^\top Q x^{(i)}\|_2 \le \|(\bar{V}_{1:i}^{(B)})^\top Q_\perp x^{(i)}\|_2, \quad \text{and} \tag{26}$$

$$\|(\bar{V}_{1:i-1}^{(B)})^\top Q x^{(i)}\|_2 = 0 \quad \text{when} \quad i \ge 2. \tag{27}$$

We further write $\hat{V}$ as a sum of a signal matrix $S$ and noise matrices $Z_1$, $Z_2$, $Z_3$, and $Z_4$ defined below.

$$
\begin{aligned}
\hat{V} &= \delta M^\top W + (\tilde{A} - \delta M)^\top W \\
&= \delta^2 \sqrt{\frac{\ell}{n}} M^\top U_{1:k'} \Sigma_{1:k'}^{1:k'} (\bar{V}_{1:k'}^{(B)})^\top Q + \delta^2 M^\top U_{1:k'} \Sigma_{1:k'}^{1:k'} (V_{1:k'}^{1:\ell} - \sqrt{\frac{\ell}{n}} \bar{V}_{1:k'}^{(B)})^\top Q + \\
&\quad \delta^2 M^\top (1 - U_{1:k'} U_{1:k'}^\top) M_{(B)} Q + \delta M^\top (A_{(B_2)} - \delta M_{(B)})Q + (\tilde{A} - \delta M)^\top W \\
&= S + Z_1 + Z_2 + Z_3 + Z_4, 
\end{aligned} \tag{28}
$$

where $S = \delta^2 \sqrt{\frac{\ell}{n}} V_{1:k'} (\Sigma_{1:k'}^{1:k'})^2 (\bar{V}_{1:k'}^{(B)})^\top Q$,

$$Z_1 = \delta^2 M^\top U_{1:k'} \Sigma_{1:k'}^{1:k'} (V_{1:k'}^{1:\ell} - \sqrt{\frac{\ell}{n}} \bar{V}_{1:k'}^{(B)})^\top Q, \qquad Z_2 = \delta^2 M^\top (1 - U_{1:k'} U_{1:k'}^\top) M_{(B)} Q,$$

$$Z_3 = \delta M^\top (A_{(B_2)} - \delta M_{(B)})Q, \qquad \text{and} \qquad Z_4 = (\tilde{A} - \delta M)^\top W.$$

Then, the signal and noise matrices amplify $x^{(i)}$ in their directions as follows:

- from (25), $\|V_i^\top S x^{(i)}\|_2 \ge \sqrt{\frac{\ell}{n}} \delta^2 s_i^2(M) \sqrt{1 - \|(\bar{V}_{1:i}^{(B)})^\top \cdot Q_\perp\|_2^2}$;

- from (26) and (27), $\|(I - V_i^\top V_i)S x^{(i)}\|_2 \le \sqrt{\frac{\ell}{n}} \delta^2 s_{i+1}^2(M) \|(\bar{V}_{1:i}^{(B)})^\top \cdot Q_\perp\|_2$;

- from (19) and $s_1(M) \le \sqrt{mn}$,

$$
\begin{aligned}
\|Z_1 x^{(i)}\|_2 &\le \delta_2 \|M^\top U_{1:k'} \Sigma_{1:k'}^{1:k'} (V_{1:k'}^{1:\ell} - \sqrt{\frac{\ell}{n}} \bar{V}_{1:k'}^{(B)})^\top\|_2 \|Q x^{(i)}\|_2 \\
&\le \delta^2 m^{2/3} \sqrt{n\ell};
\end{aligned}
$$

- from the definition of $k'$ and (20),

$$
\begin{aligned}
\|Z_2 x^{(i)}\|_2 &\le \|\delta^2 M^\top (1 - U_{1:k'} U_{1:k'}^\top)\|_2 \|(1 - U_{1:k'} U_{1:k'}^\top) M_{(B)}\|_2 \|Q x^{(i)}\|_2 \\
&\le \delta^2 s_{k'+1}(M) \sqrt{\frac{\ell}{n} s_{k+1}^2(M) + \frac{2}{\delta} \sqrt{m\ell \log(\ell)}} \\
&= \delta^2 s_{k'+1}(M) \sqrt{\frac{\ell}{n}} \sqrt{s_{k'+1}^2(M)\left(s_{k+1}^2(M) + \frac{2n}{\delta\ell}\sqrt{m\ell \log(\ell)}\right)} \\
&\le \delta^2 \sqrt{\frac{\ell}{n}} s_{k+1}^2(M) + 2\delta\sqrt{mn\log(\ell)};
\end{aligned}
$$

- from (22), $\|Z_3 x^{(i)}\|_2 = O(\delta\sqrt{mn})$;
- finally, since $\|Wx^{(i)}\|_2 = O(\sqrt{\delta m})$ thanks to the trimming process corresponding to the line 3 in the pseudo-code and $\|\tilde{A} - \delta M\|_2 = O(\sqrt{\delta(m+n)})$ from (23),

$$\|Z_4 x^{(i)}\|_2 = O(\delta\sqrt{m(m+n)}).$$

From the above conditions, when $\delta \leq m^{-8/9}$, there exists a constant $C$ such that

$$\sum_{j=1}^{4} \|Z_j x^{(i)}\|_2 \leq C\left(\delta^2\sqrt{\frac{\ell}{n}}s_{k+1}^2(M) + \delta\sqrt{mn\log(\ell)} + \delta m\right) \quad \text{and} \tag{29}$$

$$\frac{1}{4} \leq \frac{3\delta^2(s_{k+1}^2(M) + 2m^{\frac{2}{3}}n) + 3(2+c_1)\delta\frac{n}{\ell}\sqrt{m\ell\log(\ell)}}{C\left(\delta^2 s_{k+1}^2(M) + \delta\frac{n}{\ell}\sqrt{m\ell\log(\ell)} + \delta\frac{m}{\ell}\sqrt{n\ell}\right)}, \tag{30}$$

since $\frac{n}{\delta\ell}\sqrt{m\ell\log(\ell)} = n\log(m)\sqrt{\frac{m\log(\ell)}{\delta\log(m)}} > nm^{\frac{17}{18}} > nm^{\frac{2}{3}}$.

We are now ready to establish (24). When

$$\sqrt{\frac{\ell}{n}}\delta^2 s_i^2(M) \leq 2C\left(\delta^2\sqrt{\frac{\ell}{n}}s_{k+1}^2(M) + \delta\sqrt{mn\log(\ell)} + \delta m\right),$$

(24) is trivial since $\frac{\|(I-V_iV_i^\top)\hat{V}x^{(i)}\|_2}{\|\hat{V}x^{(i)}\|_2} \leq 1$ and $\frac{s_{k+1}^2(M) + n\log(m)\sqrt{m/\delta} + m\sqrt{n\log(m)/\delta}}{s_i^2(M)} = \Omega(1)$. When

$$\sqrt{\frac{\ell}{n}}\delta^2 s_i^2(M) \geq 2C\left(\delta^2\sqrt{\frac{\ell}{n}}s_{k+1}^2(M) + \delta\sqrt{mn\log(\ell)} + \delta m\right),$$

from (21), (29), and (30),

$$\|\hat{V}x^{(i)}\|_2 \geq \|V_iV_i^\top\hat{V}x^{(i)}\|_2 \geq \|V_iV_i^\top Sx^{(i)}\|_2 - \sum_{j=1}^{4}\|Z_j x^{(i)}\|_2 = \Omega\left(\sqrt{\frac{\ell}{n}}\delta^2 s_i^2(M)\right) \quad \text{and}$$

$$\|(I - V_iV_i^\top)\hat{V}x^{(i)}\|_2$$

$$\leq \|(I - V_i^\top V_i)Sx^{(i)}\|_2 + \sum_{j=1}^{4}\|Z_j x^{(i)}\|_2$$

$$\leq \sqrt{\frac{\ell}{n}}\delta^2 s_{i+1}^2(M)\|(\bar{V}_{1:i}^{(B)})^\top \cdot Q_\perp\|_2 + C\left(\delta^2\sqrt{\frac{\ell}{n}}s_{k+1}^2(M) + \delta\sqrt{mn\log(\ell)} + \delta m\right)$$

$$= O\left(\delta^2\sqrt{\frac{\ell}{n}}\left(s_{k+1}^2(M) + \frac{n}{\delta}\sqrt{\frac{m\log(\ell)}{\ell}} + \frac{m}{\delta}\sqrt{\frac{n}{\ell}}\right)\right),$$

which implies (24), since $\frac{n}{\delta}\sqrt{\frac{m\log(\ell)}{\ell}} = n\sqrt{\frac{m\log(\ell)}{\ell\delta^2}} \leq n\log(m)\sqrt{\frac{m}{\delta}}$ and $\frac{m}{\delta}\sqrt{\frac{n}{\ell}} \leq m\sqrt{\frac{n\log(m)}{\delta}}$.

## A.7 Proof of Theorem 7

The proof exploits the fact that the statements of Lemma 13 and Theorem 6 hold with probability $1 - k\delta$.

With probability $1 - k\delta$, from Theorem 6, for all $i \leq k$:

$$\|s_i(M)U_iV_i^\top(I - P_{\hat{V}})\|_F^2 = s_i(M)^2\|V_i^\top(I - P_{\hat{V}})\|_F^2$$

$$\leq s_i(M)^2\left(\min\left\{1, c_2\frac{s_{k+1}^2(M) + n\log(m)\sqrt{\frac{m}{\delta}} + m\sqrt{\frac{n\log(m)}{\delta}}}{s_i^2(M)}\right\}\right)^2$$

$$\leq c_2 \left( s_{k+1}^2(M) + n \log(m) \sqrt{\frac{m}{\delta}} + m \sqrt{\frac{n \log(m)}{\delta}} \right).$$

Therefore, since $\|M - \frac{1}{\delta}\tilde{A}\|_2 = O(\sqrt{\delta(m+n)})$ from Lemma 13,

$$
\begin{aligned}
\|M^{(k)} - \hat{U}\hat{V}^\top\|_F^2 &= \|M^{(k)} - \frac{1}{\delta}\tilde{A}P_{\hat{V}}\|_F^2 = \|M^{(k)} - MP_{\hat{V}} + (M - \frac{1}{\delta}\tilde{A})P_{\hat{V}}\|_F^2 \\
&= \|M^{(k)}(I - P_{\hat{V}}) - (M - M^{(k)})P_{\hat{V}} + (M - \frac{1}{\delta}\tilde{A})P_{\hat{V}}\|_F^2 \\
&\leq (k+2)\left( \sum_{i=1}^k \|s_i(M)U_iV_i^\top(I - P_{\hat{V}})\|_F^2 + ks_{k+1}^2(M) + k\|M - \frac{1}{\delta}\tilde{A}\|_2^2 \right) \\
&= O\left( k^2\left( s_{k+1}^2(M) + n\log(m)\sqrt{\frac{m}{\delta}} + m\sqrt{\frac{n\log(m)}{\delta}} \right) \right).
\end{aligned}
$$

## B  Matrix completion

The SLA algorithm can be extended to tackle matrix completion problems. In these problems, we wish to recover a rank-$k$ matrix $M$ from a matrix $A$ obtained by randomly erasing each entry of $M$ with probability $\delta$. We present this extended algorithm, referred to as SMC (Streaming Matrix Completion), in Algorithm 1. The only notable difference compared to SLA is that SMC has to first estimate the sampling rate $\delta$, and if the estimated sampling rate is larger than $(\log(m))^4/m$, SMC further samples the entries of $A$ so that the probability for each entry of the resulting matrix to be non-zero is equal to $(\log(m))^4/m$. We provide more explanations below.

---

**Algorithm 1** Streaming Matrix Completion (SMC)

---

**Input:** $\{A_1, \ldots, A_n\}, k$

1. $A_{(B)} \leftarrow [A_1, \ldots, A_\ell]$: store columns until the number of observed entries exceeds $\frac{m}{\log m}$.

2. $\hat{\delta} \leftarrow \frac{1}{m\ell} \sum_{(i,j)} 1([A_{(B)}]_{ij} > 0)$

3. $A_{(B)} \leftarrow$ sample $[A_{(B)}, A_{\ell+1}, A_{\lfloor \frac{m}{(\log(m))^5} \rfloor}]$ with rate $\min\{1, \frac{\log^4 m}{m\hat{\delta}}\}$

4. $A_{(B_1)}, A_{(B_2)} \leftarrow \text{Split}(A_{(B)}, 2, 2, \hat{\delta})$

5. (PCA for the first block)$Q \leftarrow \text{SPCA}(A_{(B_1)}, k)$

6. (Trimming rows and columns)

   $A_{(B_2)} \leftarrow$ make the rows having more than two observed entries to zero rows

   $A_{(B_2)} \leftarrow$ make the columns having more than $10m\hat{\delta}$ non-zero entries to zero columns

7. (Reference Columns) $W \leftarrow A_{(B_2)}Q$

8. (Principle row vectors) $\hat{V}^{1:\ell} \leftarrow (A_{(B_1)})^\top W$

9. (Principle column vectors) $\hat{I} \leftarrow A_{(B_1)}\hat{V}^{1:\ell}$

   *Remove $A_{(B)}$, $A_{(B_1)}$, $A_{(B_2)}$, and $Q$ from the memory space*

   **for** $t = \lfloor \frac{m}{(\log(m))^5} \rfloor + 1$ **to** $n$ **do**

   10. $A_t \leftarrow$ sample $A_t$ with rate $\min\{1, \frac{\log^4 m}{m\hat{\delta}}\}$

   11. $A_t \leftarrow \text{Split}(A_t, 1, 2, \hat{\delta})$

   12. (Principle row vectors) $\hat{V}^t \leftarrow (A_t)^\top W$

   13. (Principle column vectors) $\hat{I} \leftarrow \hat{I} + A_t\hat{V}^t$

   *Remove $A_t$ from the memory space*

   **end for**

14. $\hat{R} \leftarrow$ find $\hat{R}$ using the Gram-Schmidt process such that $\hat{V}\hat{R}$ is an orthonormal matrix.

15. $\hat{U} \leftarrow \frac{2}{\hat{\delta}}\hat{I}\hat{R}\hat{R}^\top$

**Output:** $|\hat{U}\hat{V}^\top|_0^1$

---

In SLA algorithm, $\delta$ is an important parameter which controls the batch size $\ell = \frac{1}{\delta \log m}$ and the output matrix $\hat{U} = \frac{1}{\delta}\hat{I}\hat{R}\hat{R}^\top$. Therefore, SMC should include some additional steps to set $\ell$ and

---

**Algorithm 2** Split

---

**Input:** $A,a,b,\delta$
**Initial:** $\tilde{A},\ldots,A^{(a)} \leftarrow$ zero matrices having the same size as $A$
**for** every $[A]_{uv}$ **do**
  $\gamma \leftarrow s \subset \{1,\ldots,b\}$ which is randomly selected over all subsets of $\{1,\ldots,b\}$ with probability
  $\frac{1}{\delta}\left(\frac{\delta}{b}\right)^{|s|}\left(1-\frac{\delta}{b}\right)^{b-|s|}$ if $s$ is not the empty set and with probability $1-\frac{1}{\delta}(1-(1-\frac{\delta}{b})^b)$ if $s$ is
  the empty set
  **for** $i \in \gamma$ **do**
    $[A^{(i)}]_{uv} \leftarrow [A]_{uv}$
  **end for**
**end for**
**Output:** $\tilde{A},\ldots,A^{(a)}$

---

compute $\frac{1}{\delta}\hat{I}\hat{R}\hat{R}^{\top}$ without prior knowledge on $\delta$. The lines 1 and 2 of Algorithm 1 are the additional part. First, in Line 1, we store columns until the number of observed entries exceeds $\frac{m}{\log m}$ so that the required memory space becomes $O(m)$. Then, in Line 2, we compute $\hat{\delta}$, an estimate of the sampling rate $\delta$. Since the number of observed entries is large enough, one can easily show Lemma 15 from the Chernoff bound.

**Lemma 15** *With probability* $1-\frac{1}{m^2}$, $\frac{|\hat{\delta}-\delta|}{\delta} = O\left(m^{-1/2}\log(m)\right)$.

Then, to handle the case where $\delta \geq \frac{\log^4 m}{m}$, the SMC algorithm undersamples the input matrix $A$ with rate $\frac{\log^4(m)}{m\hat{\delta}}$. From the undersampling process, the sampling rate of the input matrix changes to $\frac{\log^4(m)}{m}$. Since the sampling is updated to $\frac{\log^4(m)}{m}$, SMC receives more columns and stores $\frac{m}{\log^5(m)}$ columns to run the first step of the SLA algorithm.

Another non-trivial part in the SLA algorithm is the two independently sampled matrix $A_{(B_1)}$ and $A_{(B_2)}$. In Line 4 of SMC, we construct 2 undersampled copies of $A_{(B)}$ so that we have two independently sampled matrices $A_{(B_1)}$ and $A_{(B_2)}$ with rate $\frac{\delta}{2}$. Note that for the matrix completion problem, a simple undersampling procedure does not produce independently sampled matrices. The Split algorithm explains how to make such matrices and its pseudo code is give in Algorithm 2. With these additional steps, the matrix completion algorithm become similar to the SLA algorithm with sampling with rate $\frac{(\log(m))^4}{2m}$. Since $M$ is of rank $k$ (i.e., $s_{k+1}(M) = 0$), from Theorem 7 and Lemma 15, we conclude that:

$$
\begin{aligned}
\frac{\|M-[\hat{U}\hat{V}^{\top}]\|_F^2}{mn} &\leq 2\frac{\|M-\frac{\hat{\delta}}{\delta}[\hat{U}\hat{V}^{\top}]\|_F^2 + \left(1-\frac{\hat{\delta}}{\delta}\right)^2\|[\hat{U}\hat{V}^{\top}]\|_F^2}{mn} \\
&= O\left(k^2\left(\frac{\log(m)}{\sqrt{\delta m}}+\sqrt{\frac{\log(m)}{\delta n}}\right)\right) + O\left(\frac{\log(m)}{\sqrt{m}}\right) \\
&= O\left(k^2\left(\frac{1}{\log(m)}+\sqrt{\frac{m}{n(\log(m))^3}}\right)\right).
\end{aligned}
$$

**Corollary 16** *When* $\delta \geq \frac{(\log(m))^4}{m}$, *with probability* $1-k\frac{(\log(m))^4}{m}$, *the output of the SMC algorithm satisfies:*

$$
\frac{\|M-[\hat{U}\hat{V}^{\top}]_0^1\|_F^2}{mn} = O\left(k^2\left(\frac{1}{\log(m)}+\sqrt{\frac{m}{n(\log(m))^3}}\right)\right).
$$

*The complexity of SMC is* $O(kn(\log(m))^4)$ *and its memory requirement is* $O(km+kn)$.