[Reviews · NeurIPS 2015]

Submitted by Assigned_Reviewer_1

The paper proposes a new matrix approximation algorithm, which only uses a random sampled part of the matrix. After sampling the first several rows, the remaining rows come sequentially. The approximation method are space and time efficient with the same error bound as previous work.

There is two similar work. One is "Akshay Krishnamurthy, Aarti Singh. Low-Rank Matrix and Tensor Completion via Adaptive Sampling. NIPS 2013: 836-844." which is a matrix completion method sampling the columns sequentially.

Another is "Shusen Wang, Zhihua Zhang.Improving CUR matrix decomposition and the Nystroem approximation via adaptive sampling. Journal of Machine Learning Research 14(1): 2729-2769 (2013)."which adaptively sample columns/rows to approximate the matrix.

How will the results of current paper compared to the above two?
Summary: The paper proposes a new matrix approximation methods, which has three merits, 1) low time complexity, 2) low space complexity 3)same approximation error.

Submitted by Assigned_Reviewer_2

Let $M^k$ be the best rank $k$ approximation to $M$.

1. The right way to evaluating the approximation error is contrasting $|| M - \tilde{M}^k ||$ with $|| M - M^k||$, rather than contrasting $|| M^k - \tilde{M}^k ||$ to $mn$.

That $|| M^k - \tilde{M}^k || = o(mn)$ does not indicate convergence! E.g. Let m=n and $||M|| = O(n^2)$ by this paper's assumption. Assume the singular values decays slowly in that $\sigma_{max}^2 = c \sigma_{min}^2$ for a constant $c$. Then $||M^k||_F^2 < k \sigma_{max}^2 = k c \sigma_{min}^2 < k c n^{-1} ||M||_F^2 = O(kn) = o(n^2)$. This indicates that even the all-zero matrix satisfies this paper's error bound: $|| 0 - M^k ||_F^2 = o(n^2)$.

2. This paper proves an IMPOSSIBLE convergence bound. Assume that the 1st to the 2k-th singular values are equal. Any among the first 2k principal components cannot be distinguished from each other [2]. Thus $\tilde{M}^k$ can be entirely different from $M^k$, which means the impossibility of convergence. (In fact, in this case, $\tilde{M}^k$ converges to $M^k$ with ZERO measure.) Interestingly, the authors tell us their streaming PCA converges to $M^k$.

3. Stronger results have been established by Boutsidis & Woodruff [1].

[1] arXiv:1504.06729v1, April 25, 2015. Communication-optimal Distributed Principal Component Analysis in the Column-partition Model. Boutsidis, Christos, Woodruff, David P.

[2] Topics in matrix analysis. RA Horn, CR Johnson.
Summary: This paper studies the streaming PCA and offers a meaningless and impossible convergence bound. In addition, this paper overlaps Boutsidis & Woodruff [Sec 6, 1], and the result is much weaker than [1].

[1] arXiv:1504.06729v1, April 25, 2015. Communication-optimal Distributed Principal Component Analysis in the Column-partition Model. Boutsidis, Christos, Woodruff, David P.

----- Additional comments.

Admittedly, I had misunderstandings of this work due to the paper's confusing presentation. So I increase my rating.

Though the results are not trivial or wrong, the additive error bound is too weak to be meaningful. (1. Advantage over the old work of Achlioptas and Mcsherry is not enough; the stronger relative-error bounds in the literature must be compared. 2. The authors' feedback ``B. Choice of Performance Metric'' does not make any sense; otherwise any kind of bound could be turned to relative-error bound.)

Considering that 1+\eps relative-error bound has been established, I don't find the additive-error bound in this submission useful. I cannot recommend this paper for acceptance.

Author Feedback
Author rebuttal: We would like to thank the reviewers for their comments that in turn will help us to enhance our manuscript. We strongly believe that the results presented in our paper are correct (in contrast to what Reviewer 4 indicates), but understand where the confusion comes from. Our only mistake was to omit in the abstract the assumption on matrix M for our algorithm to be asymptotically accurate - this assumption is present in the introduction where results are correctly stated. The chosen performance metric was also confusing, but valid. We would also like to stress that the results of paper [1] are not stronger than ours as claimed by Reviewer 4. In addition, [1] is not published, has been posted on arxiv not long before NIPS submission deadline (we were not aware of it when submitting our paper). We also have some concerns about the results in [1] related to our problem. Next we provide more detailed arguments.

A. The notion of asymptotically accurate algorithm.

We agree with Reviewers 1 and 4 about the fact that the notion of "asymptotically accurate algorithm" as stated in the abstract does not make sense in some cases (see the example provided by Reviewers 1 and 4, where in some cases, asymptotical accuracy cannot be met). However, we were primarily interested in matrices M whose rank is approximately k, in which case the notion makes sense. Our algorithm is actually asymptotically accurate whenever $ks_{k+1}(M)=o(\qsrt{mn})$. We forgot this condition in the abstract, which led Reviewer 4 to think our results were wrong.

Note however that the actual result provided in the paper, summarized in Eq. (1), goes beyond the notion of asymptotic accuracy, and is valid for any matrix $M$.

Modifications to be made: We will remove the notion of asymptotic accuracy, and rather state it as a corollary of our result in Eq. (1).

B. Choice of Performance Metric.

In (1), we upper bound the Frobenius norm of $M^{(k)}-\hat{M}^{(k)}$ by some function, say $a$. This simply implies as stated in Eq. (2) that

(i) $\| M- \hat{M}^{(k)}\|_F \le \| M- M^{(k)}\|_F + \sqrt{a}$, and

(ii) $\| M- \hat{M}^{(k)}\|_F \le \| M- M^{(k)}\|_F (1+\epsilon)$, where $\epsilon = \sqrt{a} / \| M- M^{(k)}\|_F$.

Our result in Eq. (1) is valid for any matrix $M$ and for any rank-$k$ approximation $M^{(k)}$ (which contrasts with what Reviewer 1 indicates in her review). A bit surprisingly, Eq. (1) could be meaningful even in the example proposed by Reviewers 1 and 4 (see point 2 of Reviewer 4), where the $2k$ highest singular values are identical. In this example, we cannot design an asymptotically accurate algorithm, because $M^{(k)}$ is not unique. However in this example, we can have good performance when looking at the metrics (i) and (ii) as defined above (for (ii), this is the case when $\|M^{(k)}\|_F / \| M- M^{(k)}\|_F$ is small).

Modifications to be made: As suggested by Reviewer 1, we will keep Eq. (2) as our main result, to avoid any confusion.

C. Impossible convergence (Reviewer 4).

We believe that the confusion here originates from our choice of performance metric, as discussed in B. above. There is no contradiction in the paper. Consider the example proposed by Reviewer 4. Then in Eq. (1), we have $k^2 (s_{k+1})^2/mn$ on the right-hand side. Thus, since $s_1 = ... = s_{2k}$, Eq. (1) means $\|M^{(k)}-\hat{M}^{(k)}\|_F^2 = O(k \times \|M^{(k)}\|_F^2 + ...)$ which implies that the singular vectors of $M^{(k)}$ and $\hat{M}^{(k)}$ could be totally different. So indeed in this example, Eq. (1) still holds, and does not yield any contradiction.

D. About Boutsidis & Woodruff 2015 paper.

The two-pass streaming PCA algorithm in [1] finds k orthonormal columns U with O(mk + poly(k,\epsilon)) memory space and O(mn\log(k/\epsilon)+...) running time, whereas our SLA algorithm finds both U and V in one-pass for random input order or in two-pass streaming for any arbitrary input order with O(k(m+n)) memory space and O(\delta k mn) running time. The PCA algorithm requires one more pass to extract V from U. In addition, the running complexity of SLA to find both U and V is much lower than the running complexity of the PCA algorithm to find U. We also have a concern on Theorem 38 of [1] (the theorem stating the result for the two-pass streaming PCA). It is not clear why they do not take into account the space to store the sketching matrix W and embedding matrices T_left and T_right. If W and T are not in the memory space, they have to be generated by given seeds for every column, which increases the running time a lot more.

[1] Boutsidis & Woodruff, 25 Apr. 2015